# Sibling rank and sibling number in relation to cardiovascular disease and mortality risk: a nationwide cohort study

Peter M Nilsson [ID] ,[1] Jan Sundquist,[2] Kristina Sundquist,[2] Xinjun Li[2]

► Prepublication history and supplemental material for this paper is available online. To view these files, please visit the journal online (http://dx.doi.org/10.1136/bmjopen-2020-042881).

[1]Clinical Sciences, Lund University, Malmö, Sweden
[2]Center for Primary Health Care Research, Lund University, Malmö, Sweden

**Correspondence to**
Professor Peter M Nilsson; Peter.Nilsson@med.lu.se

## ABSTRACT

**Background** The number and rank order of siblings could be of importance for risk of cardiovascular disease and mortality. Previous studies have used only fatal events for risk prediction. We, therefore, aimed to use also non-fatal coronary and cardiovascular events in fully adjusted models.

**Methods** From the Multiple-Generation Register in Sweden, data were used from 1.36 million men and 1.32 million women (born 1932–1960), aged 30–58 years at baseline and with follow-up from 1990 to 2015. Mean age at follow-up was 67 years (range 55–83 years). Fatal and non-fatal events were retrieved from national registers.

**Results** Compared with men with no siblings, those with 1–2 siblings had a lower, and those with four or more siblings had a higher adjusted risk of cardiovascular events. Again, compared with men with no siblings, those with more than one sibling had a lower total mortality risk, and those with three or more siblings had an increased risk of coronary events.

Correspondingly, compared with women with no siblings those women with three siblings or more had an increased risk of cardiovascular events, and those with two siblings or more had an increased risk of coronary events. Women with one sibling or more were at lower total mortality risk, following full adjustment.

**Conclusion** Being first born is associated with a favourable effect on non-fatal cardiovascular and coronary events for both men and women. The underlying biological mechanisms for this should be studied in a sociocultural context.

## Strengths and limitations of this study

► This national register linkage study across generations includes data on both fatal and, for the first time, non-fatal cardiovascular and coronary events, as well as total mortality in relation to sibling number and rank.

► Adjustment has been made for confounders including markers of social background (educational level, occupation) of the individual, but not for parental socioeconomic status. Competing risk analyses have been applied.

► Limitations of the study include the lack of data from primary healthcare visits and that the historical register data do not fully reflect the ethnical diversity of Sweden today.

## INTRODUCTION

A positive family history of disease is a well-established variable to be used in risk algorithms for cardiovascular disease (CVD), even if it may be hard to quantify based on subjective recall only.[1 2] An alternative is to use register-based data for a more objective appraisal of the family burden of CVD.[3] One special feature of family structure is the number of siblings and sibling rank that can also be mapped by use of national registers such as the Multiple-Generation Register (MGR) of Sweden.[4] So far this register has

been mostly used to describe the risk of some selected CVD manifestations of individuals in relation to the risk of their siblings, for example, for thromboembolic disease[5] or other diseases.[6–9]

In a previous report based on the MGR, it was shown that increased number of sibling reflecting family size was not associated with increased total and cause-specific mortality risk in ages 40–74 years, but no analyses were made for risk of non-fatal CVD or coronary disease.[10] However, another corresponding study based on MGR data could show that total and cause-specific mortality in ages 30–69 years increased with increasing birth order.[11]

The influence of sibling rank has been less well studied in relation to non-fatal cardiovascular and coronary risk. Previous studies have indicated a worse cardiovascular risk factor burden in first borns, for example, increased body mass index (BMI) and systolic blood pressure, but lower insulin sensitivity, than later born siblings.[12–16] On the other hand, first borns seem to have a better physical fitness at military conscript testing,[17] less

caries[18] and run a lower risk of leukaemia in adolescence[19] as well as lower suicide risk in a Finnish population.[20]

Against this background we aimed to analyse the influence of sibling number and rank on risk of non-fatal and fatal cardiovascular and coronary manifestations, as well as total mortality, in the MGR of Sweden, after extensive adjustment for background demographic factors and family social status.

## SUBJECTS AND METHODS
### Setting and participants

The dataset used in this study was constructed by linking several national Swedish registers. The Swedish government-owned Statistics Sweden provided the MGR, in which persons born in Sweden in or after 1932 (the present study population) are linked to their parents.[4] We included all births (multiple births, full and half siblings) in the analyses. Linkages were made to National Census data in order to ascertain individual-level socioeconomic status. The final link was made by adding data from the Swedish Cause of Death Register (1961–2015) and the Swedish Hospital Discharge Register, with recorded dates of hospitalisation and hospital diagnoses since 1964, but on a national level since 1987 and now until end of 2015. National Swedish registers are of high validity for medical research.[21 22] For analysing risk of CVD, coronary heart disease (CHD) and mortality in relation to number of siblings and birth order, we collected data from 1.36 million men and 1.32 million women (born 1932–1960), aged 30–58 years at baseline and with follow-up from 1990 to 2015. For the definition of study subjects based on the MGR (see online supplemental figure S1).

### Patient and public involvement

No patient involved.

### Follow-up of CVD events and total mortality

We used the following International Classification of Disease (ICD)-codes for fatal or non-fatal CVD (ICD-9, 390–459, ICD-10, I00-I99); and for CHD (ICD-9, 410–414, ICD-10, I20-I25). Non-fatal events were followed in the national Hospital Discharge Register, and fatal events and total mortality until 31 December 2015 in the national Mortality Register.

### Definitions

*Family income*: family income was calculated at start of follow-up (1990) as annual family income divided by the number of members in the family, as previously reported.[23] The income calculation was weighted, taking the ages of the family members into account. For example, children were given lower consumption weights than adults. The calculation was performed as follows: the sum of all family members' incomes was multiplied by the individual's consumption weight divided by the family members' total consumption weight. The final variable was calculated as empirical

quartiles from the distribution and classified as low, middle-low, middle-high and high.

*Immigration status*: born in Sweden or in other countries.

*Marital status*: individuals were classified as married/cohabitating or never married, widowed or divorced.

*Socioeconomic status (SES)*: was divided into four categories: the self-employed/farmers/all others, blue collar workers, white collar workers or professionals, as previously reported.[24]

*Education*: was based on educational level, which was classified into three categories: ≤9 years, 10–11 years and ≥12 years.

*Geographical region*: was divided into large cities (cities with a population of more than 200 000 inhabitants), Southern Sweden and Northern Sweden.[24]

*Comorbidity*: was defined as the first hospitalisation during the follow-up period of: chronic obstructive pulmonary disease, Chronic Obstructive Pulmonary Disease (COPD) (both hospitalisation and mortality were included) (ICD-9 490–496 and ICD-10 J40-J49), obesity (ICD-9 278A and ICD-10 E65-E68), alcoholism-related liver disease (both hospitalisations and mortality were included) (ICD-9 291, 303, 571 and ICD-10 F10 and K70), hypertension (ICD-9 401–405 and ICD-10 I10–I15), and diabetes (both hospitalisations and mortality were included) (ICD-9 250 and ICD-10 E10-E14) and cancer (cancer were included both from cancer register and mortality, ICD-9 140–239 and ICD-10 C00-D48).

### Statistical methods

Person-years at risk were calculated from the start of follow-up on 1 January 1990 until hospitalisation or death from CVD, death from other causes, emigration or the end of the follow-up, 31 December 2015. Age-adjusted incidence rates for first hospitalisation and mortality were calculated for the entire follow-up period. We used the Cox's proportional hazard model to calculate the HR with 95% CIs for total (fatal and non-fatal) CVD and CHD event risk, and for total mortality, for both men and women) in relation to number siblings and birth order. This was done after adjustment for age at start, individual characteristics (family income, marital status, immigrant background and educational level, region of residence, socioeconomic status) and finally for comorbidities in order to adjust for competing mortality risk. Individuals without sibling was used as the reference. The proportionality assumptions were checked by plotting the incidence rates over time and by calculating Schoenfeld (partial) residuals and these assumptions were fulfilled. We used SAS V.9.4 (SAS Institute) for all statistical analyses.

A further adjustment was made for total number of siblings in relation to birth order when the risk for different outcomes was calculated, using the category 'first birth' as reference. A competing risk model used for mortality as a competing risk for incident CVD. A p<0.05 was considered significant.

**Table 1** Distribution of population, number of CVD, CHD and mortality events

| | Population | | CVD events | | CHD events | | Mortality events | |
|---|---|---|---|---|---|---|---|---|
| | No | (%) | No (% of population) | % | No (% of population) | % | No (% of population) | % |
| **Men** | 1 358 647 | | 592 863 (43.6) | | 131 533 (9.7) | | 240 371 (17.7) | |
| No of sibling | | | | | | | | |
| Non-sibling | 214 700 | 15.8 | 105 516 (49.1) | 17.8 | 23 671 (11.0) | 18 | 50 709 (23.6) | 21.1 |
| One sibling | 443 877 | 32.7 | 189 839 (42.8) | 32 | 39 729 (9.0) | 30.2 | 73 140 (16.5) | 30.4 |
| Two siblings | 338 812 | 24.9 | 140 361 (41.4) | 23.7 | 30 184 (8.9) | 22.9 | 52 790 (15.6) | 22 |
| Three siblings | 183 067 | 13.5 | 77 378 (42.3) | 13.1 | 17 663 (9.6) | 13.4 | 30 266 (16.5) | 12.6 |
| Four or more siblings | 178 191 | 13.1 | 79 769 (44.8) | 13.5 | 20 286 (11.4) | 15.4 | 33 466 (18.8) | 13.9 |
| Birth order | | | | | | | | |
| First | 684 765 | 50.4 | 318 341 (46.5) | 53.7 | 70 238 (10.3) | 53.4 | 140 857 (20.6) | 58.6 |
| Second | 402 879 | 29.7 | 166 757 (41.4) | 28.1 | 36 654 (9.1) | 27.9 | 62 267 (15.5) | 25.9 |
| Third | 164 540 | 12.1 | 65 853 (40.0) | 11.1 | 14 736 (9.0) | 11.2 | 23 081 (14.0) | 9.6 |
| Fourth | 62 765 | 4.6 | 24 729 (39.4) | 4.2 | 5737 (9.1) | 4.4 | 8425 (13.4) | 3.5 |
| Fifth+ | 43 698 | 3.2 | 17 183 (39.3) | 2.9 | 4168 (9.5) | 3.2 | 5741 (13.1) | 2.4 |
| **Women** | 1 315 037 | | 486 147 (37.0) | | 55 933 (4.3) | | 160 269 (12.2) | |
| No of sibling | | | | | | | | |
| Non-sibling | 210 121 | 16 | 87 261 (41.5) | 17.9 | 10 289 (4.9) | 18.4 | 34 521 (16.4) | 21.5 |
| One sibling | 430 315 | 32.7 | 154 154 (35.8) | 31.7 | 16 280 (3.8) | 29.1 | 49 132 (11.4) | 30.7 |
| Two siblings | 324 379 | 24.7 | 113 739 (35.1) | 23.4 | 12 500 (3.9) | 22.3 | 34 843 (10.7) | 21.7 |
| Three siblings | 176 631 | 13.4 | 63 871 (36.2) | 13.1 | 7512 (4.3) | 13.4 | 19 766 (11.2) | 12.3 |
| Four or more siblings | 173 591 | 13.2 | 67 122 (38.7) | 13.8 | 9352 (5.4) | 16.7 | 22 007 (12.7) | 13.7 |
| Birth order | | | | | | | | |
| First | 664 459 | 50.5 | 262 015 (39.4) | 53.9 | 30 342 (4.6) | 54.2 | 94 779 (14.3) | 59.1 |
| Second | 388 391 | 29.5 | 136 263 (35.1) | 28 | 15 191 (3.9) | 27.2 | 40 984 (10.6) | 25.6 |
| Third | 159 311 | 12.1 | 53 711 (33.7) | 11 | 6231 (3.9) | 11.1 | 15 205 (9.5) | 9.5 |
| Fourth | 60 676 | 4.6 | 20 264 (33.4) | 4.2 | 2379 (3.9) | 4.3 | 5558 (9.2) | 3.5 |
| Fifth+ | 42 200 | 3.2 | 13 894 (32.9) | 2.9 | 1790 (4.2) | 3.2 | 3743 (8.9) | 2.3 |

CHD, coronary heart disease; CVD, cardiovascular disease.

## RESULTS

With an average of 20 years (Q1–Q3 16–25 years) follow-up, in 1 358 647 men we used data on 592 863 CVD events, 131 533 coronary events and 240 371 total deaths. For 1 315 037 women, the corresponding numbers were 486 147 CVD events, 55 933 coronary events and 160 269 deaths, respectively. The mean age of the study population at the end of the follow-up was 67 years (range 55–83 years). The number of siblings and birth order of men and women are depicted in table 1.

### Risk associated with number of siblings in men and women

Compared with men with no siblings, those with 1–2 siblings had a lower, and those with four or more siblings had a higher risk of cardiovascular events. Again, compared with men with no siblings, those with more than one sibling had a lower total mortality risk, and those with three or more siblings had an increased risk of coronary events, following full adjustment (table 2).

Correspondingly, compared with women with no siblings those women with three siblings or more had an increased risk of cardiovascular events, and those with two siblings or more had an increased risk of coronary events. Women with one sibling or more were at lower total mortality risk, following full adjustment, table 3.

### Risk associated with sibling rank in men and women

According to sibling rank, first-born men had a lower risk of both cardiovascular and coronary events than their later-born siblings, but higher total mortality than second and third-born siblings, following full adjustment (table 2).

For first-born women the risks of cardiovascular and coronary events were also lower than in their later-born siblings. The mortality risk was higher than for second-born siblings, but equal to higher numbered siblings, following full adjustment (table 3).

**Table 2** HR and 95% CI of CVD, CHD and mortality in men

| | CVD | | CHD | | Mortality | |
|---|---|---|---|---|---|---|
| | HR*† | 95% CI | HR* | 95% CI | HR* | 95% CI |
| No of siblings (ref. Non sibling) | | | | | | |
| One sibling | 0.98 | 0.97 to 0.99 | 0.99 | 0.97 to 1.01 | 0.93 | 0.92 to 0.94 |
| Two siblings | 0.97 | 0.97 to 0.98 | 1.01 | 0.99 to 1.03 | 0.91 | 0.9 to 0.92 |
| Three siblings | 0.98 | 0.97 to 0.99 | 1.04 | 1.02 to 1.07 | 0.93 | 0.92 to 0.94 |
| Four or more siblings | 1 | 0.99 to 1.01 | 1.1 | 1.07 to 1.12 | 0.96 | 0.94 to 0.97 |
| Birth order (ref. first birth) | | | | | | |
| Second | 1 | 1 to 1.01 | 1.08 | 1.06 to 1.09 | 0.96 | 0.95 to 0.97 |
| Third | 1.02 | 1.02 to 1.03 | 1.13 | 1.11 to 1.15 | 0.98 | 0.96 to 0.99 |
| Fourth | 1.04 | 1.02 to 1.05 | 1.17 | 1.14 to 1.21 | 0.98 | 0.95 to 1 |
| Fifth+ | 1.07 | 1.05 to 1.09 | 1.23 | 1.19 to 1.28 | 1.01 | 0.98 to 1.05 |

*Full adjusted model: Adjusted for age at start, individual characteristics of family income, marital status, educational attainment, immigrant status, socioeconomic status, region of residence, comorbidities, number of siblings and birth order.
†Multivariable competing risk survival analysis.
CHD, coronary heart disease; CVD, cardiovascular disease.

For HRs of CHD, CVD and total mortality by number of siblings and birth order in men and women, respectively (see figures 1–3).

### Supplemental material

For detailed data on the distribution of the study population, number of CVD, CHD and mortality events in men and women (see online supplemental table S1). For detailed data on the risk associated with factors adjusted (see online supplemental tables S2–S4), for men and women, respectively.

### DISCUSSION

In this very large observational study based on a national MGR, it was found that first-born men and women are at lower risk of both cardiovascular and coronary events than their later-born siblings, but had higher total mortality risk than second and third-born siblings (men). For women the mortality risk for first-born women was higher than for second-born siblings, following full adjustment for a number of background factors.

For total mortality in relation to sibling number our data are at odds with a previous study using the same register in Sweden, showing no increased mortality associated until 74 years with a higher number of siblings.[10] However, we used higher numbers, longer follow-up and more extensive adjustment.

For the influence of sibling rank, a previous study could show higher risk of total and cause-specific mortality with increasing sibling rank until 69 years.[11] This was similar in

**Table 3** HR and 95% CI of CVD, CHD and mortality in women

| | CVD | | CHD | | Mortality | |
|---|---|---|---|---|---|---|
| | HR*† | 95% CI | HR* | 95% CI | HR* | 95% CI |
| No of siblings (ref. non-sibling) | | | | | | |
| One sibling | 0.98 | 0.98 to 0.99 | 0.99 | 0.97 to 1.02 | 0.94 | 0.93 to 0.95 |
| Two siblings | 0.99 | 0.98 to 1 | 1.03 | 1 to 1.06 | 0.92 | 0.91 to 0.94 |
| Three siblings | 1 | 0.99 to 1.01 | 1.07 | 1.04 to 1.11 | 0.93 | 0.91 to 0.95 |
| Four or more siblings | 1.01 | 1 to 1.03 | 1.17 | 1.13 to 1.21 | 0.95 | 0.93 to 0.96 |
| Birth order (ref. first birth) | | | | | | |
| Second | 1.01 | 1 to 1.02 | 1.07 | 1.05 to 1.09 | 0.96 | 0.95 to 0.98 |
| Third | 1.02 | 1.01 to 1.03 | 1.14 | 1.11 to 1.18 | 0.98 | 0.96 to 1 |
| Fourth | 1.04 | 1.02 to 1.05 | 1.14 | 1.09 to 1.2 | 1 | 0.98 to 1.03 |
| Fifth+ | 1.05 | 1.03 to 1.07 | 1.22 | 1.15 to 1.29 | 1.03 | 0.99 to 1.07 |

*Full adjusted model: Adjusted for age at start, individual characteristics of family income, marital status, educational attainment, immigrant status, socioeconomic status, region of residence, comorbidities, number of siblings, and birth order.
†Multivariable competing risk survival analysis.
CHD, coronary heart disease; CVD, cardiovascular disease.

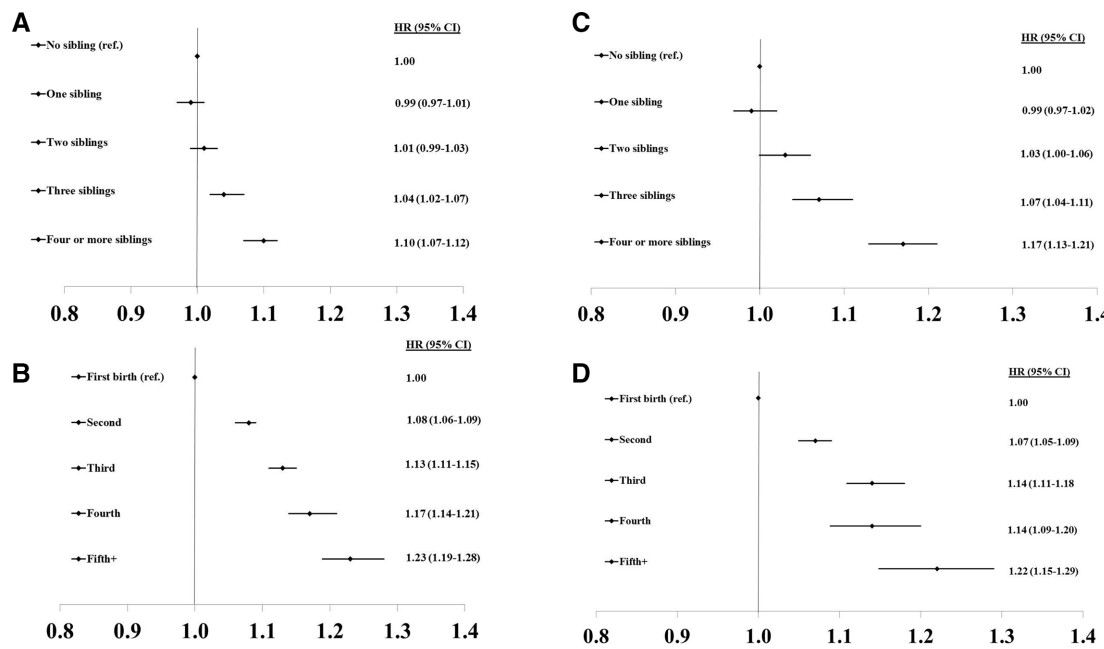

**Figure 1** HRs of CHD by number of siblings and birth order in men (A, B) and women (C, D). CHD, coronary heart disease.

our study for risk of non-fatal cardiovascular and coronary events during longer follow-up and extensive adjustment.

These findings of lower cardiovascular risk in first borns are at contrast to previous reports of a higher level of cardiovascular risk factors in such individuals followed until adolescence or young adulthood.[12–16] The burden of risk factors might have been compensated for by a better physical fitness, as noticed in first-born men coming for military conscript testing at the age of around 18 years.[17] In contrast to these observations, our extensive data

indicate a lower cardiovascular risk in first borns. Other unmeasured factors linked to being first born, such as cognition or bodily development, could have contributed to our findings of a relative protection, even if we adjusted for a long list of potential confounders such as educational level, socioeconomic status, marital status and comorbidities.

Besides filling a knowledge gap, this is of public health interest as different countries endorse different policies to support families and number of children. Our findings

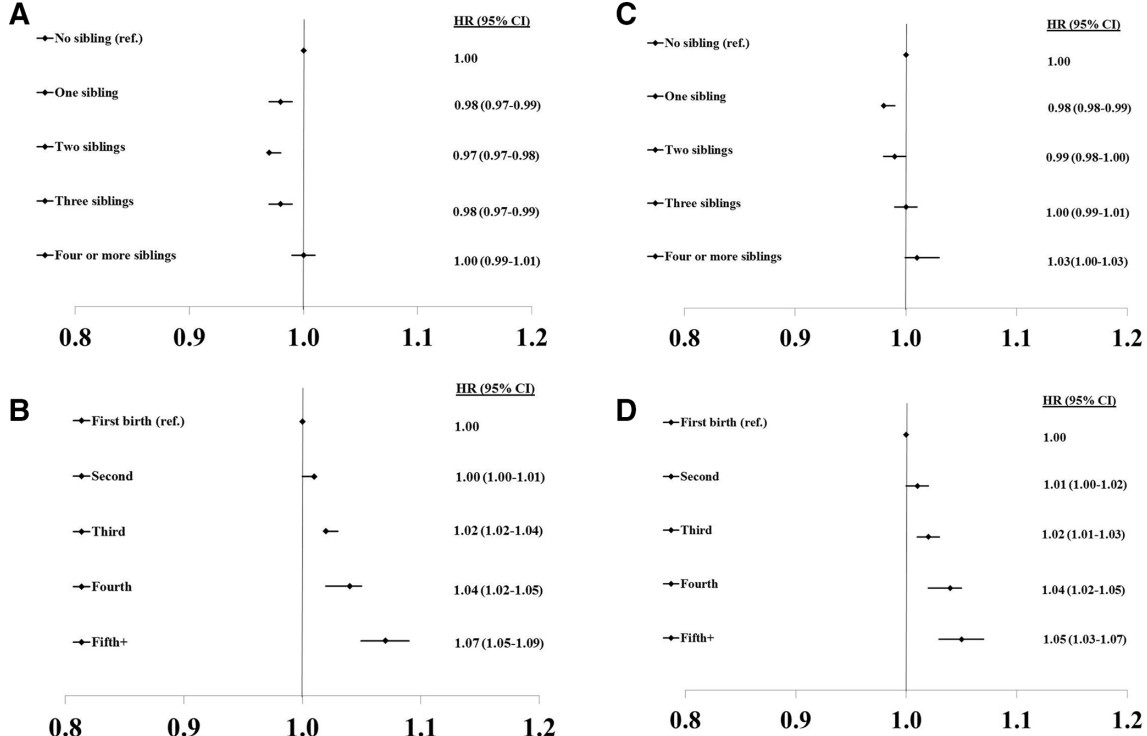

**Figure 2** HRs of CVD by number of siblings and birth order in men (A, B) and women (C, D). CVD, cardiovascular disease.

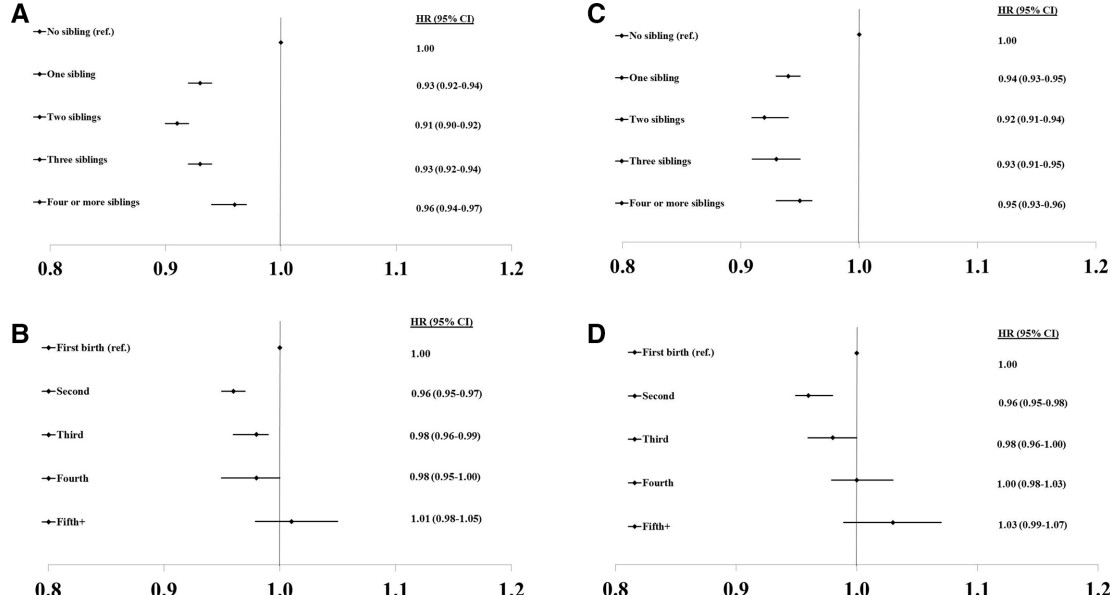

**Figure 3** HRs of mortality by number of siblings and birth order in men (A, B) and women (C, D).

relate to family size and the biological as well as social roles related to sibling rank with its health implications.

More research is needed to understand the links between sibling number and rank with health outcomes. This could address, for example, the dilution of resources theory[25] of special relevance for disadvantaged girls; epigenetic factors influencing the metabolic syndrome in offspring[26] and maternal health during pregnancy, including the effects of multiple births/child rearing on maternal health and family resources, especially in deprived settings with large families.[27]

### Limitations and strengths of the study

The Swedish hospital discharge register contains no information about diagnostic procedures, which is a limitation. Moreover, specialist doctors in hospital care made the diagnosis. Another limitation is that we had no data on life style-related factors such as BMI, smoking and diet, because it would be unrealistic to gather such data for an entire national population. However, we did adjust for socioeconomic status, obesity, diabetes, COPD and alcoholism and related liver disorders, which are associated with factors such as smoking and alcohol use. Given a focus on family size, knowing that siblings who died young, and therefore not contributed to resource dilution for a proportion of the index person's childhood, would be of interest and importance. It would also shed light (potentially) on family circumstances and health. Regretfully, we currently lack data on parental SES to adjust for.

Strengths of the study include complete nationwide coverage from 1990 in a country with high standards of diagnosis, and with diagnoses often being made by specialists during extended examinations in clinics. Another important strength of our study is that it was based on nationwide registers and was thus free of selection and recall bias. The Swedish MGR and the Swedish

Hospital Discharge Register are validated data sources that have been proven to be reliable in the study of many diseases.[4 21 22] Data in our dataset are almost 100% complete.[4] Generalisability (external validity) should hold at least for countries and populations similar to Sweden.

Future research should be directed to find biological or social mechanisms linking the status of being first born to lower risk of CVD, as indicated by our observational findings. A previous Norwegian study in military conscripts indicated that the role of being first born is influenced by social factors, as a second-born son may achieve characteristics of a first-born brother who died young.[28]

In conclusion, our data indicate a favourable effect on non-fatal cardiovascular and coronary events by being first born, both for men and women.

**Contributors** PMN provided the original idea, and XL made the statistical analyses. PMN and XL drafted the first manuscript. All authors (PMN, XL, JS and KS) contributed to the final manuscript. PMN and XL made the revisions.

**Funding** This work was supported by the grants to Kristina Sundquist from The Swedish Research Council; ALF funding from Region Skåne and the Swedish Heart-Lung foundation.

**Competing interests** None declared.

**Patient consent for publication** Not required.

**Ethics approval** The study was approved by the Regional Ethics Review Board at the Lund University, Sweden (ID number 2012/795; approved 2013-02-06).

**Provenance and peer review** Not commissioned; externally peer reviewed.

**Data availability statement** No data are available. Data will not be shared, but reasonable requests may be directed to Professor KS, the PI of the registers.

for any error and/or omissions arising from translation and adaptation or otherwise.

**ORCID iD**
Peter M Nilsson http://orcid.org/0000-0002-5652-8459

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
