## [Reviewer comments · BMJ Open]

ARTICLE DETAILS

TITLE (PROVISIONAL)	Sibling rank and sibling number in relation to cardiovascular disease and mortality risk: a nationwide cohort study
AUTHORS	Nilsson, Peter; Sundquist, Jan; Sundquist, Kristina; Li, Xinjun

VERSION 1 – REVIEW

REVIEWER	Dr Claire E Welsh Newcastle University
REVIEW RETURNED	13-Nov-2020

GENERAL COMMENTS	Thank you for the chance to review this interesting manuscript. The study is impressive in size and very well written, I do, however, have a number of major questions about omissions of the analysis and methodological approaches which would need to be addressed before I could advise publication. Major points: 1. It is unclear whether siblings who died before the start of follow-up would be included in the analysis. Given a focus on family size, knowing that siblings had died young, and therefore not contributed to resource dilution for a proportion of the index person's childhood, would be of interest and importance. It would also shed light (potentially) on family circumstances and health.2. No mention is made of how multiple births were dealt with, nor whether only full siblings or half siblings also were included. The sex mix of the siblings is also ignored, but all of these factors are used and discussed in a number of the references used in the manuscript. Why were they ignored here?3. More clarity is required around the covariates – and missing but important covariates such as the parental (or maternal) SES/wealth and age at the time of birth are important, but omitted. Why? If they are not available then a frank discussion of how this may have affected the results is needed.4. Sibling rank and number are unlikely to be independent of each other in their associations with later-life health. I agree with stratifying the analyses, but I think it would be important to adjust for rank when analysing number and vice versa, plus to report interaction effects, neither of which appear to have been done.5. Diagnoses at death were available in the dataset used. It would be beneficial to summarise the differences in causes of death between first-born and later-born children, for example, to shed light on possible mechanisms by which mortality effects varied.6. Much more discussion is needed around the possible mechanisms underlying the results. Also, filling a knowledge gap is helpful, but the aims of the work beyond that are unclear – what would you propose is done with these results? Should it inform public health policy in some way, or is it purely of scientific interest?
---

	7. Cox models may not be the best approach here. Given that the data can identify family groups, some estimation of shared frailty within a family should be incorporated, in a hierarchical structure. Competing risks models for mortality as a competing risk for CVD should perhaps have been considered also. If simple multivariable Cox models were still thought to be the best approach, justification needs to be made. Additional Points: Page 5, Line 28: More information (possibly in the supplement) is required to explain and reference the weighting method used. Unequal division of family funds between adult family members is a known issue – how was this handled or why was it not adjusted for? Page 5, Line 38: Socioeconomic status. Why was a validated index not used? If this is a standard way to measure SES then references are needed. If this is a bespoke method then further explanation of how job families were categorised is required. Surely unemployed and retired people should be separate from all other self-employed people? Any, please provide references or explanations. Page 5, Line 54: Why was the Cause of Death register not also used to classify the other comorbidities that surely lead to a proportion of the deaths e.g. alcohol related liver disease, diabetes? Page 6, Line 31: IQR is a single number, either report it or state 'Q1-Q3' instead of 'IQR' Statistical methods: A number of important steps in the analysis have not been reported. Did you check the proportional hazards assumption, how, and what was the result, any estimates of model goodness-of-fit? Were any interactions tested for? What about checking the univariable associations of each covariate with the outcome? Your results report three incremental models – please describe this approach here (as it's written, the reader expects a single, fully-adjusted model) Table 1: It would be useful to also report the % of CVD/CHD/mortality as the number of cases/number in that group, e.g. None sibling CVD would be 105516/214700. Table 2: Please list all the adjustments made. Discussion: Much more is needed here discussing possible mechanisms by which the results have come about, many have been mentioned above, for example the dilution of resources theory, epigenetic factors and maternal health during pregnancy, including the effects of multiple births/child-rearing on maternal health and family resources.
--	--

VERSION 1 – AUTHOR RESPONSE

Reviewer: 1

Dr. Claire Welsh, Newcastle University

Competing interests of Reviewer: None declared

Comments to the Author:

Thank you for the chance to review this interesting manuscript. The study is impressive in size and very well written, I do, however, have a number of major questions about omissions of the analysis and methodological approaches which would need to be addressed before I could advise publication.

Major points:

1. It is unclear whether siblings who died before the start of follow-up would be included in the analysis. Given a focus on family size, knowing that siblings had died young, and therefore not contributed to resource dilution for a proportion of the index person's childhood, would be of interest and importance. It would also shed light (potentially) on family circumstances and health.

Author response: We agree with the reviewer's comment, however our study was focused on the effects related to risks of CVD/CHD and mortality by number of siblings and birth order.

We could add the sentence in the Discussion (page 8) as a limitation:

"Given a focus on family size, knowing that siblings who died young, and therefore not contributed to resource dilution for a proportion of the index person's childhood, would be of interest and importance. It would also shed light (potentially) on family circumstances and health".

2. No mention is made of how multiple births were dealt with, nor whether only full siblings or half siblings also were included. The sex mix of the siblings is also ignored, but all of these factors are used and discussed in a number of the references used in the manuscript. Why were they ignored here?

Author response: We have now added this information and mentioned that we included all births (multiple births, full and half siblings) in the analyses (page 4).

3. More clarity is required around the covariates – and missing but important covariates such as the parental (or maternal) SES/wealth and age at the time of birth are important, but omitted. Why? If they are not available then a frank discussion of how this may have affected the results is needed.

Author response: Unfortunately we do not have access to this exposure information on parental/maternal factors of importance such as SES. This is now mentioned in the Discussion (page 8) as a limitation.

4. Sibling rank and number are unlikely to be independent of each other in their associations with later-life health. I agree with stratifying the analyses, but I think it would be important to adjust for rank when analysing number and vice versa, plus to report interaction effects, neither of which appear to have been done.

Author response: Thank you for your important comment, we have now adjusted for sibling rank when analyzing numbers of siblings and vice versa. In addition, we have tested interaction effects between number of siblings and individual characteristics. There was no meaningful interaction shown except for "number of siblings" and "immigrants (foreign-born) status". Immigrants with few siblings show a negative (protective) effect on CVD risk as compared to Swedish-born individuals, see Table below.

Table. Correlation of individual factors and CVD

Born in Sweden Foreign-born

Number of siblings HR* 95% CI HR* 95% CI

No sibling 1.00 0.94 0.91 0.98

One sibling 0.99 0.98 1.00 0.94 0.92 0.97

Two siblings 0.99 0.99 1.00 0.94 0.92 0.97

Three siblings 1.00 0.99 1.01 1.01 0.98 1.05

Four or more siblings 1.02 1.01 1.03 1.03 1.00 1.06

*: Fully adjusted.

5. Diagnoses at death were available in the dataset used. It would be beneficial to summarise the differences in causes of death between first-born and later-born children, for example, to shed light on possible mechanisms by which mortality effects varied.

Author response: We have checked the distribution of causes of death between first-born and later-

born children. The cause of death due to cardiovascular disease accounted for 26.5% in the first-born and 21.6% in the later-born children ($p < 0.001$). However, there was no difference by cause of death due to any cancer, 2.1% vs. 2.2%, respectively ($p = 0.77$).

6. Much more discussion is needed around the possible mechanisms underlying the results. Also, filling a knowledge gap is helpful, but the aims of the work beyond that are unclear – what would you propose is done with these results? Should it inform public health policy in some way, or is it purely of scientific interest?

Author response: Thank you for your comments, we have added below sentences in the Discussion, and also contributed text on mechanisms (page 8):

“Besides filling a knowledge gap, this is of public health interest as different countries endorse different policies to support families and number of children. Our findings relate to family size and the biological as well as social roles related to sibling rank with health implications.

More research is needed to understand the links between sibling number and rank with health outcomes. This could address, for example, the dilution of resources theory [23] of special relevance for girls, epigenetic factors influencing the metabolic syndrome in offspring [24], and maternal health during pregnancy, including the effects of multiple births/child-rearing on maternal health and family resources, especially in deprived settings with large families [25]”.

7. Cox models may not be the best approach here. Given that the data can identify family groups, some estimation of shared frailty within a family should be incorporated, in a hierarchical structure. Competing risks models for mortality as a competing risk for CVD should perhaps have been considered also. If simple multivariable Cox models were still thought to be the best approach, justification needs to be made.

Author response: Thanks for this important suggestion, we have done the requested analyses and included a competing risks model for mortality as a competing risk for incident CVD.

Additional Points:

Page 5, Line 28: More information (possibly in the supplement) is required to explain and reference the weighting method used. Unequal division of family funds between adult family members is a known issue – how was this handled or why was it not adjusted for?

Author response: Thank you for your comments, we have clarified the variable of family income (page 5), as follows:

“Family income: Family income was calculated at start of follow-up (1990) as annual family income divided by the number of members in the family. The income calculation was weighted, taking the ages of the family members into account. For example, children were given lower consumption weights than adults. The calculation was performed as follows: the sum of all family members’ incomes was multiplied by the individual’s consumption weight divided by the family members’ total consumption weight. The final variable was calculated as empirical quartiles from the distribution and classified as low, middle-low, middle-high, and high”.

Page 5, Line 38: Socioeconomic status. Why was a validated index not used? If this is a standard way to measure SES then references are needed. If this is a bespoke method then further explanation of how job families were categorised is required. Surely unemployed and retired people should be separate from all other self-employed people? Any, please provide references or explanations.

Author response: The method of classification of socioeconomic status (SES) applied is based on what is used in Swedish national statistics and register research (Ludvigsson JF, et al. Registers of the Swedish total population and their use in medical research. *Eur J Epidemiol.* 2016; 31(2):125-36). This method and indices have previously been used by many authors for epidemiological studies based on national data from Sweden. Educational level is another measure of SES that we have

used. This means that we have two valid measures of SES used for adjustment, even if some sort of misclassification may still exist.

Page 5, Line 54: Why was the Cause of Death register not also used to classify the other comorbidities that surely lead to a proportion of the deaths e.g. alcohol related liver disease, diabetes?

Author response: Thank you for your suggestion, we have now added (included) cause of death caused by comorbidities caused by diabetes, alcoholism and related liver disease.

Page 6, Line 31: IQR is a single number, either report it or state 'Q1-Q3' instead of 'IQR'

Author response: We have changed it.

Statistical methods: A number of important steps in the analysis have not been reported. Did you check the proportional hazards assumption, how, and what was the result, any estimates of model goodness-of-fit? Were any interactions tested for? What about checking the univariable associations of each covariate with the outcome? Your results report three incremental models – please describe this approach here (as it's written, the reader expects a single, fully-adjusted model)

Author response: We have now done the test for the proportional hazards assumption, and added a sentence in the Methods (page 6):

"The proportionality assumptions were checked by plotting the incidence rates over time and by calculating Schoenfeld (partial) residuals and these assumptions were fulfilled. We used SAS version 9.4 (SAS Institute Inc. Cary, NC, USA) for all statistical analyses".

Table 1: It would be useful to also report the % of CVD/CHD/mortality as the number of cases/number in that group, e.g. none sibling CVD would be 105516/214700.

Author response: Thank you for your suggestion, Table 1 has now been changed and proportions were added as requested.

Table 2: Please list all the adjustments made.

Author response: A list of all the adjustments was added for each Table.

Discussion: Much more is needed here discussing possible mechanisms by which the results have come about, many have been mentioned above, for example the dilution of resources theory, epigenetic factors and maternal health during pregnancy, including the effects of multiple births/child-rearing on maternal health and family resources.

Author response: We have now expanded the Discussion on these relevant aspects of possible mechanisms linking sibling number and rank to health outcomes (page 8):

"Besides filling a knowledge gap, this is of public health interest as different countries endorse different policies to support families and number of children. Our findings relate to family size and the biological as well as social roles related to sibling rank with health implications.

More research is needed to understand the links between sibling number and rank with health outcomes. This could address, for example, the dilution of resources theory [23] of special relevance for girls, epigenetic factors influencing the metabolic syndrome in offspring [24], and maternal health during pregnancy, including the effects of multiple births/child-rearing on maternal health and family resources, especially in deprived settings with large families [25]".